# Analysis of Aroma Characteristics of ‘Binzi’ and ‘Xiangguo’ Apple—Ancient Cultivars in China

**DOI:** 10.3390/foods13182869

**Published:** 2024-09-10

**Authors:** Xiang Lu, Zhao Liu, Yuan Gao, Kun Wang, Simiao Sun, Hanxin Guo, Wen Tian, Lin Wang, Zichen Li, Lianwen Li, Jianrong Feng, Dajiang Wang

**Affiliations:** 1Xinjiang Production and Construction Corps Key Laboratory of Special Fruits and Vegetables Cultivation Physiology and Germplasm Resources Utilization, College of Agriculture, Shihezi University, Shihezi 832000, China; xianglu1997@163.com (X.L.); lz__0427@163.com (Z.L.); tianwen8025@163.com (W.T.); 2Research Institute of Pomology, Chinese Academy of Agricultural Sciences/Key Laboratory of Horticultural Crops Germplasm Resources Utilization, Ministry of Agriculture and Rural Affairs, Xingcheng 125100, China; gaoyuan02@caas.cn (Y.G.); wangkun@caas.cn (K.W.); sunsimiao@caas.cn (S.S.); 82101235244@caas.cn (H.G.); juziwanglin@163.com (L.W.); lishencheno@163.com (Z.L.); lilianwen@caas.cn (L.L.)

**Keywords:** Binzi, Xiangguo, fruit aroma, storage, characteristics aroma, metabolism pathway

## Abstract

‘Binzi’ (BZ) (*Malus domestica* subsp. *chinensis* var. *binzi* Li Y.N.) and ‘Xiangguo’ (XG) (*Malus domestica* subsp. *chinensis* var. *xiangguo* Li Y.N.) are the ancient cultivars in China. The BZ fruits have a low-fragrant flavor on harvest day but a high-fragrant flavor after storage at room temperature, while the XG fruits have a stronger flavor when mature. ‘Starking’ (SK) and ‘Golden Delicious’ (GD) fruits have a rich flavor and are recognized by all countries in the world. However, information on the differences between ancient Chinese cultivars and Western apple cultivars in aroma compounds remains unknown. The apple fruits were collected for continuous two years. Aroma compounds in the skin and pulp of the fruits were detected at room temperature (20 ± 1 °C) during storage. The dynamics of VOCs in BZ and SK fruits were more similarly reflected in esters, while those of XG and GD fruits were reflected in aldehydes and alcohols. Ethyl 2-methylbutyrate, with an extremely low odor threshold, was the main source of typical apple flavor in SK, BZ, and XG fruits, while hexyl acetate was the source of the banana flavor in GD fruits. 6-methyl-5-hepten-2-one and β-damascenone were the important ketones produced in the later stage of storage, derived from the carotenoid metabolism pathway and providing a citrus and rose flavor to the four apple cultivars. SK had the highest number of characteristic aroma components, which were mainly derived from the amino acid metabolism pathway, providing fruits with a sweet and fruity flavor. Although the characteristic aroma components of GD were derived from the fatty acid metabolic pathway, the number of volatile esters was lower. Ethyl butyrate, derived from the saturated fatty acid metabolism, had the highest content in BZ, providing a pineapple flavor; the flavor of XG was mainly derived from ethyl 2-methylbutyrate, 6-methyl-5-hepten-2-one, and β-damascenone. Therefore, we suggest BZ and XG apples as the aroma-breeding material with which to enrich new cultivars’ aroma components, derived from the fatty acid metabolism and carotenoid metabolism pathways, respectively.

## 1. Introduction

Binzi (*Malus domestica* subsp. *chinensis* var. *binzi* Li Y.N.) and Xiangguo (*Malus domestica* subsp. *chinensis* var. *xianggo* Li. Y.N.), belonging to the *Malus domestica* Borkh species, are the ancient cultivars in Hebei, Beijing, Shaanxi, and Shanxi, probably from a natural hybrid of Mianpingguo (*Malus domestica* subsp. *Chinensis* Li. Y.N.) and Shaguo (*Malus asiatica* Nakai) [1]. At present, Binzi and Xiangguo are widely planted in Huailai and Zhuolu county, Hebei Province [2]. It is reported that ‘Binzi’ includes nearly 20 varieties, such as ‘Binqiu’, ‘Huangbinzi’, ‘Xiangbinzi’, ‘Yuanbinzi’, ‘Hutoubin’, ‘Binzi’, etc. Xiangguo included ‘Matixiangguo’, ‘Pingpo’, ‘Qiutianguo’, and ‘Xiangguo’ [3]. Significantly, we discovered that ‘Binzi’ was low-fragrant at harvest day and high-fragrant after storage at room temperature, while ‘Xiangguo’ has a strong fragrance when it is mature. 

Aroma is a complex mixture of a large number of volatile organic compounds (VOCs) formed from esters, aldehydes, alcohols, ketones, acids, alkenes, and hydrocarbons [4]. Each VOC has a unique odor, and their combinations, concentrations, and ratios result in fruits with unique aroma characteristics [5]. More than 300 volatile compounds have been reported in apples, but only 20–30 compounds with low odor thresholds are significant contributors to the typical apple aroma [6,7]. Aldehydes, esters, and alcohols are the main VOCs playing an important role in immature and mature apples, respectively [8,9,10]. Hexanal, butanal, trans-2-hexena, butyl acetate, hexyl acetate, ethyl 2-methyl-butanoate, 1-propanol, 1-butanol, and 1-hexanol are the important volatile components affecting the fragrance of different apple cultivars [11,12]. Ethyl butanoate, ethyl 2-methylbutanoate, and 2-methylbutyl acetate have been identified as important contributors to the flavor of the ‘Pink Lady’ apple [13]. Hexyl 2-methylbutyrate, α-farnesene, and (E)-2-hexenal have been found to contribute significantly to the flavor of the ‘Honeycrisp’ apple [14]. Hexanal and (E)-2-hexenal are vital compounds in the flavor of the ‘Ruixue’ apple, which are responsible for its grassy and green notes [15]. Butyl acetate and hexyl acetate are present in nearly 515 apple varieties [16]. However, information on the VOCs of ancient cultivars in China is unknown. Therefore, it is of great significance to investigate of the volatile profiles of ancient cultivars in China.

‘Starking’ and ‘Golden Delicious’ apples are the main parents of apple cultivars in apple breeding. ‘Starking’ is bred via ‘Delicious’ through bud sport. ‘Golden Delicious’ is a yellow skin cultivar with a certain banana flavor after harvest; hence, it is also called ‘Yellow Banana’ in China [3]. A series of apple cultivars belonging to the Delicious and Golden Delicious strains have been bred using ‘Starking’ and ‘Golden Delicious’ apples, which have similar aroma profiles (Delicious strains belong to rich fragrance types and Golden Delicious strains belong to green fragrance types) [17,18]. Therefore, it is urgent to improve the diversity of the apple’s aroma profile to meet consumer and market demands. Hybrid breeding can increase the types and content of aroma components [19]. Further, the cultivation of apple fruit traits requires that the apples go through the juvenile stage before evaluation, which is a huge challenge for apple aroma breeding [20]. Finding specific aromatic apple resources is a shortcut for breeding. However, the differences in aroma components between ancient, intensely flavored cultivars in China (‘Binzi’ and ‘Xiangguo’) and internationally renowned strong aroma cultivars (‘Starking’ and ‘Golden Delicious’) are unknown, which prevents the utilization of special cultivars and the process of aroma breeding. Therefore, exploring apple resources with unique aromas is of great significance for aroma breeding.

Gas chromatography coupled to mass spectrometry (GC-MS) is commonly used for detecting volatile compounds. Solid phase microextraction (SPME) is a method for extracting VOCs from food which is sensitive, efficient, and low-cost. SPME-GC-MS is widely used to detect fruit aroma and pesticide residue [21,22,23]. 

This study detected the VOCs of ‘Binzi’, ‘Xiangguo’, ‘Starking’, and ‘Golden Delicious’ apple fruits during storage for two consecutive years. The main VOCs stably present in apples were identified by comparing the differences in two years. Principal component analysis (PCA) and orthogonal partial least square-discriminant analysis (OPLS-DA) were used to analyze the differences in VOCs among different cultivars during post-harvest storage, clarifying the differences between ancient cultivated apples in China and mainstream cultivated apples. Based on the olfactory characteristics and thresholds of different VOCs, the characteristic aroma compounds and fruit flavor among different cultivars were identified. We analyzed the differences in the quantity and content of VOCs from different aroma metabolism pathways and clarified the reasons for the differences in aroma among cultivars. This will provide a theoretical basis for apple species utilization and apple aroma breeding. 

## 2. Materials and Methods

### 2.1. Plant Materials and Post-Harvest Storage Conditions

The fruits of ‘Starking’ (SK) and ‘Golden Delicious’ (GD) were collected from the demonstration garden of the Research Institute of Pomology of Chinese Academy of Agricultural Sciences, Xingcheng city, Liaoning province, China (120°44′ E, 40°37′ N) for two consecutive years. The fruits of ‘Xiangguo’ (XG) and ‘Binzi’ (BZ) were collected from a village in Huailai county, Hebei province, China (115°31′ E, 40°24′ N). Three trees that arrived at the full bearing period were randomly selected. Mature fruits of uniform size and without visible damage were selected and transported to the laboratory and storage at 20 ± 1 °C until losing edible value. 

At all stages, the samples were carefully dissected from the skin or pulp during storage. The first stage was the harvest day (S1); the second stage (S2) was 5 days after S1; the third stage (S3) was 10 days after S1; the fourth stage was 20 days after S1; the fifth stage (S5) was 30 days after S1. According to the edibility of fruits during storage, we selected five stages of SK, four stages of GD, three stages of BZ, and two stages of XG. All samples were cut into pieces and treated with liquid nitrogen and stored at −80 °C until VOCs detection.

### 2.2. Preparation of Apple Fruit Samples before VOC Detection

After storage was completed, the fruit samples were homogenized, and 10.0 g of each sample was weighed and placed in an injection bottle with 3.2 g of NaCl (purity > 99%; Sangon Biotech, Shanghai, China); then, 22 μL of the internal standard (2-octanol, 0.5 g·L^−1^) was added, mixed well, and then covered.

### 2.3. VOC of Apple Fruits Using Solid-Phase Microextraction, Gas Chromatography and Mass Spectrometry (SPME-GC-MS)

We took out the preserved samples and enriched the VOC with SPME; then, were set up the GC-MS conditions to test the VOC. The detailed SPME method and the conditions of GC-MS were described in our previous study [24].

### 2.4. Identification and Quantitation of VOCs

The total ion flow chromatogram (TIC) was recorded in full scanning mode. The components of each chromatographic peak were identified by computer for matches in two mass spectrometry libraries (NIST and Wiley) and combined with artificial atlas and data. The mass spectra of all components under each chromatographic peak were compared with the standard mass spectra in the libraries to obtain similarity (SI) and reverse similarity (RSI), which were used to identify the component of each chromatographic peak. SI and RSI > 700 were the lowest reference standards. Meanwhile, previous studies on the composition of fruits have been reported as auxiliary references. The content of each VOC was determined via the internal standard method. The detailed methods of quantitation could be found in our previous study [24].

### 2.5. Calculation of Odor Activity Value (OAV)

OAV was used to describe the contribution of each compound to the overall flavor. The aroma description and odor threshold of each compound can be found in previous reports [25,26] and the following websites: https://www.flavornet.org/flavornet.html (accessed 22 June 2024); http://www.leffingwell.com/odorthre.htm (accessed 7 September 2024); and https://www.femaflavor.org/flavor-library (accessed 7 September 2024). The OAV of each VOC represents the ratio of its content to the odor threshold. The formula of OAV [27] is as follows: $${OAV}_{i}=\frac{C_{i}}{{OT}_{i}}.$$

C_i_ and OT_i_ represent the concentration and order threshold of the target compound, respectively.

### 2.6. Data Analysis 

Data were presented as the mean values of three biological replicates. The upset plot, Venn diagram, and radar map were constructed using the Hiplot (https://hiplot.com.cn/home/index.html, accessed 22 June 2024). The Origin 2021b software was used to conduct principal component analysis (PCA). OPLS-DA analysis was performed using the OmicShare tools (https://www.omicshare.com/tools, accessed 22 June 2024). The heatmap was drawn by TBtools [28].

## 3. Results

### 3.1. Identification and Determination of VOCs in Four Apple Cultivars

VOCs of SK, GD, BZ, and XG fruits in maturity and regular temperature storage periods were extracted and detected using SPME-GC-MS. A total of 105 VOCs were detected, including 55 esters, 16 aldehydes, 13 alcohols, 7 ketones, 3 acids, 3 terpenes, and 8 others (Appendix A). As shown in Figure 1A–D, SK fruit had the highest number of VOCs, totaling 89 VOCs, followed by GD fruit with 71 VOCs, BZ fruit with 53 VOCs, and XG fruit with 49 VOCs. Esters, aldehydes, alcohols, and ketones were the most numerous aroma categories in the four apple cultivars, accounting for over 80% of the total aroma components. 

The compositions of VOCs among the four apple fruits are shown in Figure 1A–D. The esters had the largest difference in number, ranging from 17 to 47 in the four apple cultivars; while the difference in the number of other aroma categories was less than 5. The results indicated that esters might be the main compounds causing differences in the aroma of the four apple cultivars.

A total of 28 common VOCs were shared in 4 apple cultivars, and 11, 5, 5, and 2 unique VOCs were only detected in SK, BZ, GD, and XG fruits, respectively (Figure 1E). These common VOCs and unique VOCs might be the compounds responsible for differences in fruit flavor. According to the upset plot, the compositions of VOCs in sets ‘A’, ‘D’, ‘G’, ‘H’, and ‘K’ are shown in Figure 1F. Esters had the greatest number in common and unique VOCs. The unique VOCs in SK fruit were propyl propionate, propyl tiglate, 2-methylbutylbutyrate, isoamyl butyrate, amyl 2-methyl butyrate, hexyl butyrate, isobutyl hexanoate, 2-methylbutylpentanoate, octanoic acid propyl ester, 1-methylheptyl acetate, and 4-methyl-1-pentanol; the unique VOCs in GD fruit were amyl butyrate, octyl formate, hexyl isobutyrate, ethyl trans-4-decenoate, and 1,3-octadiene; the unique VOCs in BZ fruit were ethyl propionate, ethyl 3-hydroxybutyrate, ethyl sorbate, 2,3-butanediol, and hexane; the unique VOCs in XG fruit were 2-methyl-4-pentanolide and (2Z)-2-hexen-1-ol.

### 3.2. Core VOCs of Four Apple Cultivars in Two Years

Environmental factors from different years might have a significant impact on the VOCs of apples. The number of VOCs was significantly varied in different years (Figure 2). And 59, 46, 30, and 31 core VOCs were shared in two years for SK, GD, BZ, and XG fruits, respectively (Figure 2A–D). Further analysis was conducted on the relative content of these core VOCs in different years, as shown in Figure 2E–H. Whether in the first or second year, the relative content of core VOCs accounted for over 80% of the total content, even exceeding 90% in SK, GD, and XG fruits. These core VOCs were the main and stable compounds in the four apple cultivars. To study the aroma differences of four apple cultivars, the following will analyze these core VOCs. 

### 3.3. Aroma Profile of Four Apple Cultivars during Storage

#### 3.3.1. Changes in Total Aroma Content

The changes in total aroma content are shown in Figure 3A. At the period of fruit maturity, the total content of SK fruit was significantly higher than that of GD, BZ, and XG fruits. In the first year, the total aroma content of SK and GD fruits reached their peak at S3, while the total aroma content of BZ and XG fruits reached their peak in S2. In the second year, the total aroma content of SK, GD, and BZ fruits reached their peak at S4, S3, and S3, respectively; while the total aroma content of XG fruit changed relatively little. Based on the two-year testing results, it can be concluded that the total aroma content growth period of SK and GD fruits occurred from S2 to S3, and BZ fruit mainly occurred from S1 to S2, while the change in XG fruit was smaller.

#### 3.3.2. Difference of Aroma Categories in Different Fruit Tissues

There were obvious differences in the aroma components and contents between the skin and pulp among cultivars during the post-harvest stages. The content in the skin was significantly greater than that in the pulp, and the categories of VOCs were quite different (Figure 3B–I).

In the skin, esters were the primary compounds that increased most in SK, GD, and BZ fruits during storage, while ketones were the compounds that increased most in XG fruit. (Figure 3B,D,F,H). In the first year of fruit ripening (S1), the highest ester content was found in SK fruit; esters and aldehydes were close to each other in GD fruit; while alcohols predominated in BZ fruit; and aldehydes, ketones, and esters were similar in XG fruit. The difference between the second and first year was mainly caused by the change in ester content, which was higher in SK, GD, and BZ fruit and lower in XG fruit compared with the first year. With the increase in fruit storage time, the main compounds that increased in SK fruit were esters; in GD fruit, the content of esters increased, while the content of aldehydes and alcohols decreased; in BZ fruit, the main compounds that increased were esters and ketones; the content of XG fruit changed less, showing a slight increase in ketones and a decrease in aldehydes.

In the pulp, SK fruit were mainly composed of esters, GD and BZ fruit were mainly composed of esters and alcohols, and XG fruit were mainly alcohols (Figure 3C,E,G,I). At fruit ripening (S1), the predominant aroma compounds in SK and GD fruits were esters, and the predominant compounds in BZ and XG fruits were alcohols. With the storage of fruit, the esters content increased rapidly in SK and BZ fruits, whereas the esters and alcohols content increased simultaneously in GD fruit; XG fruit showed a rapid increase in alcohols in the first year and less change in the second year. 

### 3.4. PCA of Four Apple Cultivars during Storage

PCA revealed the differences of different post-harvest stages on the VOCs. The differences of four apple cultivars are shown in Appendix A. SK and GD fruits, with the furthest distributed on the score plot, were able to represent two different flavor types; while XG and BZ fruits were less different from each other. Notably, BZ fruit was closer to SK fruit on the score plot, while XG fruit was closer to GD fruit. 

Each apple cultivar was analyzed via PCA, and the results are shown in Figure 4. The apple samples between two years had consistency. The fruits in different tissues could be separated by PC1, and in different storage periods, they could be separated by PC2. A critical time point could distinguish the mature fruits and aroma transition fruits at the apple storage: S3 was the critical time point for SK and GD fruits; and S2 was the critical time point for BZ fruit. XG fruit had significant differences between the S1 and S2. Combined with the loading plot, there was a significant difference in the VOCs among cultivars at the aroma transition period. In the skin, SK fruit was associated with butyl hexanoate, 2-methyl-butanoic acid butyl ester, 6-methyl-5-hepten-2-one, ethyl 2-methylbutyrate, and propyl 2-methylbutyrate, etc.; GD fruit was associated with (E)-2-octenal, 1-nonanal, hexyl formate, 2-methyl-1-butanol, 1-butanol, hexyl alcohol, butyl hexanoate, and hexyl 2-methylbutyrate; BZ fruit was associated with 1-octen-3-ol, ethyl butyrate, ethyl caproate, ethyl acetate, ethyl 2-methylbutyrate, and hexyl alcohol; and XG fruit was associated with 2-methyl-1-butanol, hexyl alcohol, hexanal, and ethanol. In the pulp, β-damascenone was an important component that correlated significantly with the fruit storage in all cultivars.

### 3.5. OPLS-DA of Aroma Compounds during Storage

The VOCs of four apple cultivars was analyzed via OPLS-DA. Differential VOC (D-VOC) during the storage was screened during the storage based on the value of variable importance in the projection (VIP) greater than 1 and *p*-value < 0.05. A total of 45 D-VOCs were selected in the four apple cultivars (Appendix A). The amount of D-VOC was different in different years due to the environmental factors. Combining two years’ results, the key D-VOCs in different apple cultivars were singled out, which were important compounds affecting fruit flavor.

D-VOCs clustered into two categories based on changes in content during storage: the first category was produced during the pre-storage period and decreased in content during the post-storage period; while the second category was produced during the post-storage period (Appendix A). There was an intersection with high aroma content between the two categories, and the intersection was consistent with the critical time point obtained from the PCA. At the time point, 21, 16, 5, and 6 D-VOCs were screened in SK (S3), GD (S3), BZ (S2), and XG (S2) apples, respectively. Based on the content of D-VOCs, the unique D-VOCs, which were different from other cultivars, were methyl hexanoate, propyl 2-methylbutyrate, and 2-methyl butyric acid in SK fruit, hexyl hexanoate in GD fruit, ethyl butyrate and hexyl alcohol in BZ fruit, and hexanal in XG fruit, respectively.

### 3.6. Identification of the Characteristic VOCs and Flavor Profiles

The contribution of each VOC to the overall aroma of apples is determined by its OAV, which is associated not only with its concentration but also with its odor threshold. The odor threshold values for each VOC were collected from the literature and are listed in Appendix A. A total of 38 and 24 characteristic VOCs were identified in skin and pulp, respectively (Appendix A). Despite the differences in OAVs from different years, the same characteristic aroma compounds were detected, and they had similar dynamic changes, both in two years. The data in the first year (Appendix A) were more representative of the flavor profiles of the apple cultivars because their trends were more pronounced than the data in second year (Appendix A). Therefore, we used the data in the first year to explain the flavor differences among the four apple cultivars. At fruit ripening (S1), the total OAV of SK (4121.10), GD (1625.80), and XG (2253.81) fruits were higher than BZ fruit (316.69). The total OAV of SK fruit rapidly increased during the post-harvest storage, and the total OAV at S3 stage (18,366.16) was 4.98 times that of the S2 stage (3687.67), while the total OAV of the S4 stage was 2.06 times that of the S3 stage (37,995.85); the total OAV of GD (3446.87), XG (12,110.91), and BZ (31,956.86) fruits reached their peaks at S3, S2, and S2, respectively. Therefore, S3 and S2 were the important time points for significant changes in the aroma of cultivars during storage. Significantly, the time points were consistent with the critical time points previously obtained by PCA and OPLS-DA. 

At this time point, the characteristic aroma components of the skin and pulp were different. Table 1 and Table 2 showed the odor and OAV of the main characteristic VOCs in skin and pulp, respectively. As the fruits were stored, the differences among cultivars became more significant (Figure 5). In the skin, methyl 2-methylbutyrate (1679.36) was the main source of flavor for SK fruit on harvest day (S1), while ethyl 2-methylbutyrate (9268.73) was the main compound at S3. Both of them provided a typical apple flavor. 2-Methyl butyric acid (606.44) and hexyl 2-methylbutyrate (527.09), with high OAVs, endowed a banana flavor for SK fruit. Hexyl acetate (532.83), hexanal (407.84), and 1-nonanal (335.04) were the main compounds with high OAVs in GD fruit, providing a banana, grass, and citrus flavor; moreover, ethyl 2-methylbutyrate (15,469.63), ethyl butyrate (3439.60), and ethyl caproate (1167.97) had relatively high OAVs and contributed the apple and pineapple flavor to BZ fruit, making it different from other cultivars. Ethyl 2-methylbutyrate (8740.38) were the main compounds with a typical apple flavor in XG fruit. In the pulp, Ethyl 2-methylbutyrate was still the most important aromatic compound in SK (2717.36), BZ (9328.17), and XG (1036.50). β-damascenone was an important aromatic compound in the pulp of SK (769.58), GD (921.84), and XG (1574.64) fruit, providing an apple, rose, and honey flavor. In addition, ethyl butyrate was an important aromatic compound in BZ (1993.95) fruit at S2, while hexyl acetate was the main compound in SK (317.65) and GD (417.48) fruit.

### 3.7. Differences in Metabolic Pathways of Main Aroma Compounds among Cultivars

In order to understand further the formation mechanism and variation reasons for the main compounds in the four apple cultivars, the metabolic pathways have been analyzed. The main compounds were derived from fatty acid metabolism, amino acid metabolism, carotenoid metabolism, and acetaldehyde metabolism (Figure 6). 

The characteristic aroma compounds of SK fruit (2-methyl-butanoic acid butyl ester, ethyl 2-methylbutyrate, 2-methylbutyl acetate, hexyl 2-methylbutyrate, and 2-methyl butyric acid) were derived from amino acid metabolism pathways. Ethyl 2-methylbutyrate was a special compound derived from amino acid metabolism, which contributed a stronger flavor for XG and BZ fruits due to its extremely low odor threshold. The most characteristic aroma compounds of BZ, GD, and XG fruit were from fatty acid metabolism pathway. Differently, ethyl butyrate, and ethyl caproate in BZ fruit, hexanal in XG fruit, and hexyl acetate and hexyl hexanoate in GD fruit were derived from β-oxidation of saturated fatty acids metabolism pathways, while 1-nonanal and heptaldehyde in GD fruit were derived from unsaturated fatty acid metabolism pathways. In addition, the carotenoid metabolism pathway was important in all fruits, producing 6-methyl-5-hepten-2-one in the skin and β-damascenone in the pulp, which played an important role in the flavor of fruits in the later stages of storage. 

The metabolic pathways of other non-characteristic aroma components also had significant differences among cultivars. 2-Methyl-1-butanol and propyl 2-methylbutyrate were the main VOCs in SK fruit and were derived from amino acid metabolism pathways; methyl hexanoate and hexyl formate were the main VOCs in GD fruit and were derived from unsaturated fatty acid metabolism pathways; 1-butanol in GD fruit, hexyl alcohol in BZ and XG fruits, and butyl hexanoate in GD and XG fruits were derived from the β-oxidation of saturated fatty acids metabolism pathways. Ethanol derived from the acetaldehyde metabolism pathway was only detected during the later storage period of BZ and XG fruits. Although these compounds did not have an exact OAV, they produced more during storage and were probably important background substances. 

This result indicate that the VOCs in SK fruit are mainly derived from amino acid metabolism pathways; the VOCs of GD fruit mainly are derived from the fatty acid pathway; the VOCs of BZ and XG fruits are mainly derived from the β-oxidation of fatty acids metabolism pathways.

## 4. Discussion

### 4.1. The VOCs of Apples

The aroma of apples is determined by the quantity and content of VOCs. Over 300 aromatic VOCs have been identified in apple varieties [6,7]. Esters, alcohols, and aldehydes are the main aroma types in apples [29]. It has been found that the difference in the number of esters was much greater than that of aldehydes and alcohols in a large number of apple cultivars [11,12,18]. In our study, the sum of esters, aldehydes, and alcohols accounts for 80% of the total aroma amount. The number of VOCs in SK and GD fruits was much higher than that in BZ and XG fruits, and the difference in this quantity was mainly reflected in esters. For example, the number of esters ranged from 17 to 47 in the four apple cultivars; while the difference in the number of other aroma categories was smaller (<5). This result emphasized the importance of esters for apple aroma. Genetic information was an important factor affecting the VOCs among apple cultivars. SK and GD apples were native to the United States in North America, while GD and XG apples were native to China in Asia, which leads to differences in their genetic information. Therefore, hybridization can be used to integrate the genetic information of apples from different origin centers, in order to enrich the aroma components of apples.

The environmental factors on harvest day were also an important factor affecting fruit VOCs [30]. Nevertheless, due to environmental differences between different years, there were still differences in the VOCs. We compared the VOCs between two years and found that the same VOCs in two years only account for about 56.6%~66.3% of the total amount. Interestingly, the content of these same VOCs account for 81.2%~96.8% of the total content. This indicates that although the environment of different years affected the numbers of apple aroma compounds, the relative content of main components did not undergo significant changes. The difference in the number and relative content of VOCs in SK and GD fruits between two years was smaller than that in BZ and XG fruits. Heavy rains dilute flavor compounds in tomatoes before harvest, and mild water deficit supply can increase grape aroma content [31]. BZ and XG apples were in the rainy season when they mature (late August), while SK and GD apples were in the sunny season (early October), which might result in a greater difference in relative content between two years of BZ and XG apples than that of SK and GD apples.

### 4.2. Dynamic Changes of VOCs during Storage Period

As typical climacteric fruits, the post-harvest storage stage was an important period for apples to produce a large amount of aroma [14,32]. A previous study showed that the concentration of apple aroma compounds increased during ripening and reached the maximum at the climacteric stage [6]. Most VOCs (esters, alcohols, sesquiterpenes) showed maxima at the ‘Honeycrisp’ apple’s climacteric peak [14]. ‘Nanguo’ pear produced a large number of esters at 12 days after harvest [33]. ‘Rocha’ pear also had a stronger flavor at 20 °C compared to 10 °C and 1-methylcyclopropene treatment [34]. In this study, most aromas in SK and GD apples were produced 10 days after harvest; the aroma content of BZ apple increased rapidly at 5 days after harvest; while the total aroma content of XG apple changed little. This reflected the influence of apple cultivars characteristics on its aroma. Ethylene played a vital role in promoting metabolic processes, many of which may contribute to the synthesis of substrates used in the formation of esters, but ethylene accelerated the aging of the fruit and reduced the storage time [35,36,37]. BZ and XG apples had poor storage performance with the fruit aging, which probably led to their aroma peaks reaching earlier compared with SK and GD apples. Wang et al. [38] also proved that ‘Xiangbinzi’ and ‘Hulabin’ apples produced more ethylene and esters during storage.

Principal component analysis (PCA) is a useful statistical tool that can reduce the dimensions of large datasets and reveal the possible differences between samples [39,40]. OPLS-DA was a supervised dimension reduction method, which could identify difficult-to-find and complex variables [41]. Combining PCA and OPLS-DA, we distinguished the aroma characteristics of different cultivars and screened for key differentially expressed VOCs. The differences in VOCs among the four varieties are mainly reflected in the skins. The optimal aroma periods for SK, GD, and BZ apples were S3, S3, and S2, respectively; while the aroma changes of XG during S1 and S2 were relatively small. Through OPLS-DA, we screened the unique VOCs differentially during the storage period: methyl hexanoate, propyl 2-methylbutyrate, and 2-methyl butyric acid in SK fruit; hexyl hexanoate in GD fruit; ethyl butyrate and hexyl alcohol in BZ fruit; and hexanal in XG fruit. The 6-methyl-5-hepten-2-one was the common components in four apple cultivars with a significant change in content at storage. In our previous study, 6-methyl-5-hepten-2-one was detected in all 50 apple cultivars, but the difference in its content was small among cultivars [18]. Significantly, hexyl 2-methylbutyrate was the most potent aroma component, followed by ethyl acetate, ethyl butyrate, (E)-2-hexenal, and Hexyl alcohol in ‘Xiangbinzi’ fruit [38]; while ethyl butyrate was the main compound in BZ fruit, followed by 6-methyl-5-hepten-2-one and ethyl caproate. This result indicated that the BZ apple used in this experiment differed from the ‘Xiangbinzi’ apple. 

### 4.3. Characteristic VOCs and Flavor Profiles of Apples

Among the more than 300 reported VOCs, only 20–30 compounds were the main contributors to the typical apple aroma [7]. Odor activity value (OAV), defined as the ratio of the concentrations of VOCs to their odor threshold values in water [42], and was an indicator of the contribution of individual VOC to the aroma of foods [43]. The contribution of each VOC to the overall aroma of apples is determined by its OAV, which was associated with its concentration and odor threshold [44]. For example, butyl acetate, hexyl acetate, 2-methylbutyl acetate, and ethyl 2-methylbutanoate had a low aroma threshold value and were important odor-active compounds in ‘Honeycrisp’ apple [14]. In this study, we identified 42 VOCs as characteristic aroma components due to their OAVs exceeding 1. 2-Methyl butyric acid, ethyl 2-methylbutyrate, 2-methylbutyl acetate, hexyl 2-methylbutyrate, ethyl caproate, ethyl butyrate, hexyl acetate, hexanal, 1-nonanal, and β-damascenone were the main contributors to the aroma of the four apple cultivars. In the fruit mature period, the Delicious strain apples were often described as sweet and fruity fruits, while the Golden Delicious strain apples were described as green fragrant fruits [18]. In this study, the aroma of fruits varied significantly at different storage stages. At harvest day, 2-methyl butyric acid, ethyl 2-methylbutyrate, 2-methylbutyl acetate, ethyl butyrate, hexyl 2-methylbutyrate, ethyl caproate, and hexyl acetate endowed the SK fruit with a apple, pineapple, fruity, and pungent flavors, resulting a sweet and fruity flavor; while hexanal, 1-nonanal, and hexyl acetate endowed GD fruit with grass, citrus, banana, and fruity flavors, resulting a green fragrant flavor; ethyl 2-methylbutyrate, hexanal, and 1-nonanal endowed XG fruit with apple, citrus, pepper, grass, and fruity flavors, with its aroma possessing SK and GD apples characteristics; while BZ fruit had almost no fragrance at the harvest day. When the fruit reached its key aroma transition period, ethyl 2-methylbutyrate and ethyl butyrate occupied an absolute position in the flavor of BZ, endowing it with a typical apple and pineapple flavor and significantly distinguishing it from other apples. Due to the production of 2-methylbutyl acetate and hexyl acetate, there was a certain banana and fruity flavor in SK and GD fruits, and hexanal and 1-nonanal also enhanced the green aroma of GD fruit to a certain extent. Under the interaction of VOCs, banana flavor may exhibit stronger performance in GD fruit than in SK fruit. Therefore, post-harvest storage made the flavor of SK fruit more intense; while GD fruit exhibited a banana flavor, combined with its yellow skin characteristics, giving it the name of ‘yellow banana’. Notably, β-damascenone was an important aromatic compound in the pulp, providing a flavor of apple, rose, and honey to SK, GD, and XG fruits. 

### 4.4. Metabolism Pathway of VOCs in Apples

The main aroma metabolism pathways in apples included fatty acid metabolism, amino acid metabolism, carotenoid metabolism, terpenoids metabolism, and acetaldehyde metabolism. The aroma components of volatile esters come from two precursors: fatty acids and amino acids [45]. Fatty acid metabolism was the main pathway for producing linear esters, while amino acid metabolism was the main pathway for producing branched esters [46]. Characteristic aroma components determined the flavor profile of the apple. 2-Methyl-butanoic acid butyl ester, ethyl 2-methylbutyrate, 2-methylbutyl acetate, hexyl 2-methylbutyrate, and 2-methyl butyric acid were the main characteristic aroma components in SK fruit and were derived from amino acid metabolism pathways; while the other characteristic aroma compounds in BZ, GD, and XG fruit were mainly derived from fatty acid metabolism pathway. It suggested that the differences in fruit flavor between the SK fruit and the other three apple cultivars were due to the different metabolic pathways. Meanwhile, some non-characteristic aroma components with higher content differences during storage, namely, 2-methyl-1-butanol and propyl 2-methylbutyrate, in the SK fruit were derived from amino acid metabolism pathways, which meant that the SK fruit was more active in amino acid metabolic pathways. 

Fatty acid metabolism was further divided into saturated fatty acid metabolism and unsaturated fatty acid metabolism [46]. β-oxidation was the primary pathway for fatty acid synthesis, and it yielded alcohols and acyl-coenzyme A (acyl-CoA) for ester formation. Through β-oxidation, short-chain fatty acids such as hexanoic acid, butyric acid, and acetic acid were produced. These acids were reduced to their corresponding alcohols, which finally form esters under the action of acyl-CoA and alcohol acyltransferase (AAT) [8,47,48]. In the lipoxygenase (LOX) pathway, lipoprotein-associated phospholipase 2 (PLA2G) can hydrolyze phospholipids to produce unsaturated fatty acids, while LOX converts unsaturated fatty acids into hydroperoxides. Subsequently, the corresponding C6 and C9 aldehydes were generated through the action of hydroperoxide lyase (HPL), oxidation, cleavage, and dehydrogenation [49]. The last important step in the biosynthesis of volatile esters was the esterification of alcohol with acetyl CoA by catalyzing different alcohol acyltransferases (AAT) [50]. In this study, hexyl alcohol, ethyl butyrate, and ethyl caproate in BZ apple, hexanal, butyl hexanoate, and hexyl alcohol in XG apple were derived from β-oxidation pathways. The characteristic aroma compounds in GD fruit were derived from the β-oxidation pathway (hexyl formate, hexanal, and methyl hexanoate) and the unsaturated fatty acid metabolism pathway (1-butanol, butyl hexanoate, and hexyl hexanoate), and some no-characteristic aroma compounds with higher content were also derived from the unsaturated fatty acid metabolism pathway. Differently, these VOCs in XG and GD fruits were mainly volatile aldehydes and alcohols, while in BZ fruit, they were mainly volatile esters. This difference probably was related to AAT. Ethyl butyrate was the most abundant compound produced in the β-oxidation pathway and was reported to have a strong correlation with Acyl-CoA synthetase (ACSL) [51]. ACSL played a vital role in fatty acid oxidation by catalyzing the production of acyl-CoA, which activated β-oxidation [46,52]. Thus, ACSL activity may be much stronger in BZ apples than in XG and GD apples. In addition, ethanol was only detected during the storage period of BZ and XG apples. Ethanol steam treatment catalyzed the fabrication of ester acyl CoA and precursor alcohols, thereby enhancing the volatile esters’ release from ‘Fuji’ apples [53]. The VOCs of BZ and XG fruits mainly came from the β-oxidation of the saturated fatty acids pathway, and the acyl CoA was the rate-limiting step in the fatty acid metabolism pathway. This suggested that ethanol might indirectly promote the production of VOCs derived from the fatty acid metabolism pathways in BZ and XG fruits. 

Carotenoid metabolism was the major metabolic pathway for the production of volatile ketones [54]. Geranyl acetone, 6-methyl-5-hepten-2-one, β-ionone, and β-damascenone were the main VOCs produced in the carotenoid metabolism pathway. 6-Methyl-5-hepten-2-one, geranyl acetone, and β-ionone have been detected in tomatoes, apples, and melons [51,55]. β-Damascenone played an important role in the flavor of wine and was also detected in apples [18,56]; due to its lower taste threshold, it contributed significantly to the aroma of apples. In this study, 6-methyl-5-hepten-2-one and β-damascenone were the most important aromatic components, making significant contributions to fruit flavor in the later stages of storage. 6-methyl-5-hepten-2-one in SK, GD, and BZ fruits was mainly produced at late stage of fruit storage, while the content of 6-methyl-5-hepten-2-one in XG fruits was higher at harvest day. This indicated that the activity of the carotenoid metabolism pathway in XG fruit was stronger than that of the others at the early stages of fruit storage.

## 5. Conclusions

In this study, we investigated the volatile profiles of two ancient cultivars in China (‘Binzi’ and ‘Xiangguo’ apples) and two internationally renowned apple cultivars (‘Starking’ and ‘Golden Delicious’ apples). Esters were the main VOCs in ‘Starking’, ‘Golden Delicious’, and ‘Binzi’ fruit, while Ketones and aldehydes were the main VOCs in the ‘Xiangguo’ fruit. Through two consecutive years of data, stable and high-content compounds were screened out. Forty-five D-VOCs were screened out by OPLS-DA. A total of 42 VOCs were identified as characteristic aroma components. At fruit ripening, ‘Binzi’ fruit had the lowest OAV; while at the optimal stage, BZ had the highest OAV. 2-Methyl-butanoic acid butyl ester, ethyl 2-methylbutyrate, 2-methylbutyl acetate, and hexyl 2-methylbutyrate were the main characteristic aroma components in SK fruit and were derived from the amino acid metabolism pathway, providing fruits with a sweet and fruity flavor. Hexanal, 1-nonanal, hexyl 2-methylbutyrate, and hexyl acetate were the main characteristic aroma components in ‘Golden Delicious’ fruits derived from the fatty acid metabolic pathway, providing fresh, grass, and banana flavors. The characteristic aroma components of ‘Binzi’ and ‘Xiangguo’ fruit were mainly derived from the β-oxidation of saturated fatty acid metabolic pathways. Ethyl 2-methylbutyrate, ethyl butyrate, and ethyl caproate was the main components of ‘Binzi’ fruit, providing pineapple and apple flavors; the flavor of ‘Xiangguo’ apple was mainly derived from ethyl 2-methylbutyrate, producing an apple flavor. In addition, the carotenoid metabolism pathway was an important element of all apples, producing 6-methyl-5-hepten-2-one in the skin and β-damascenone in the pulp, which plays an important role in the flavor of apples in the later stages of storage. The main reason for the flavor differences between Chinese ancient cultivars and western apple varieties is reflected in metabolic pathways. ‘Starking’ fruit had more compounds derived from the amino acid metabolism; ‘Golden Delicious’ fruit contained more compounds derived from the fatty acid metabolism pathway but lacked esters from the fatty acid metabolism pathway. ‘Binzi’ fruit produced more esters from the saturated fatty acids metabolism pathway. ‘Xiangguo’ apple produced more ketones from the carotenoid metabolism pathway at harvest day. Therefore, we suggested that the ‘Binzi’ apple be selected as a material for aroma breeding to increase the amount and content of volatile esters originating from the β-oxidation pathway of the saturated fatty acid metabolism pathway; the ‘Xiangguo’ apple can be selected as a material for aroma breeding to increase the content of volatile ketones originating from the carotenoid metabolism pathway at fruit ripening. 

## Figures and Tables

**Figure 1 foods-13-02869-f001:**
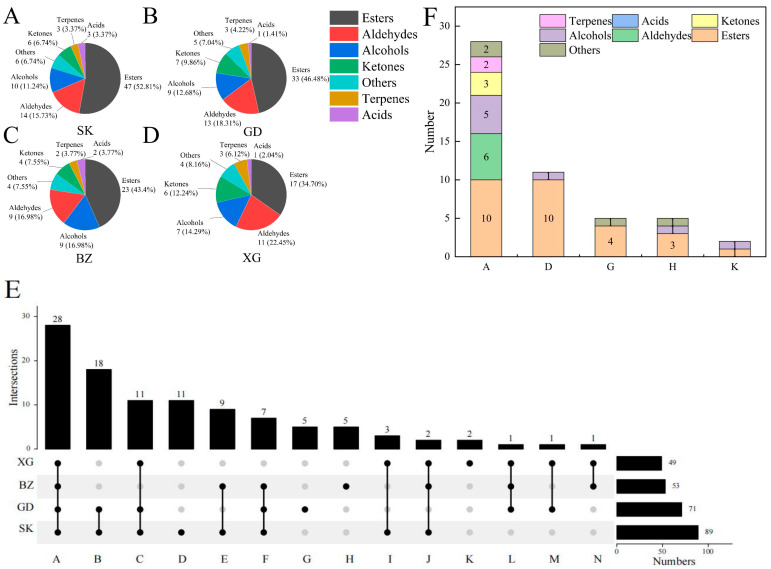
Composition of VOCs in SK (**A**), GD (**B**), BZ (**C**), and XG (**D**). (**E**) Upset plot of four apple cultivars. Set ‘A’ represents the common VOCs among the four apple cultivars; set ‘B’ represents the common VOCs between SK and GD; set ‘C’ represents the common VOCs among SK, GD, and XG; set ‘E’ represents the common VOCs between SK and BZ; set ‘F’ represents the common VOCs among SK, GD, and BZ; set ‘I’ represents the common VOCs between SK and XG; set ‘J’ represents the common VOCs among SK, BZ, and XG; set ‘L’ represents the common VOCs among GD, BZ, and XG; set ‘M’ represents the common VOCs between GD and XG; set ‘N’ represents the common VOCs between BZ and XG; and set ‘D’, ‘G’, ‘H’, and ‘K’ represents the unique VOCs in SK, GD, BZ, and XG, respectively. (**F**) The composition of VOCs of set ‘A’, ‘D’, ‘G’, ‘H’, and ‘K’ from in (**E**).

**Figure 2 foods-13-02869-f002:**
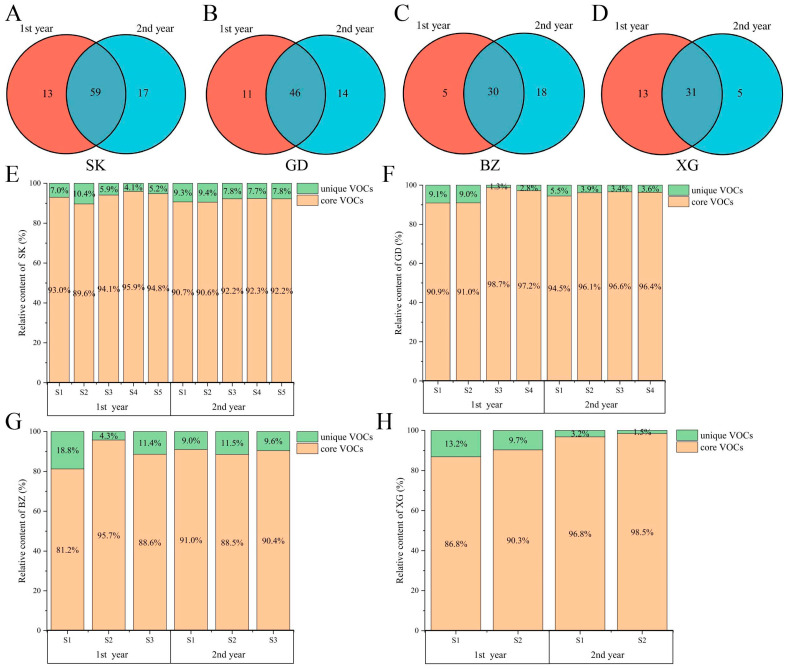
Venn plots of VOCs from different years in SK (**A**), GD (**B**), BZ (**C**), and XG (**D**). Percentage bar chart of aroma content in SK (**E**), GD (**F**), BZ (**G**), and XG (**H**).

**Figure 3 foods-13-02869-f003:**
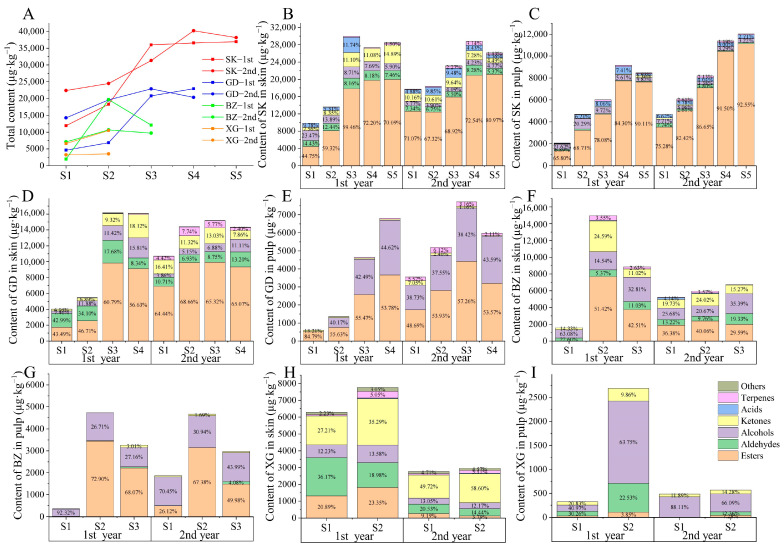
Dynamic changes in aroma content of four apple cultivars during storage period. (**A**) Dynamic changes in total content of SK, GD, BZ, and XG during storage period. 1st and 2nd represent the first year and second year, respectively. Aroma component content in (**B**) SK, (**D**) GD, (**F**) BZ, and (**H**) XG fruit skins. Aroma component content in (**C**) SK, (**E**) GD, (**G**) BZ, and (**I**) XG fruit pulps. Percentage represents the relative content of each aroma categories to the total content. The percentage less than 1% in Figure (**B**–**I**) has been hidden.

**Figure 4 foods-13-02869-f004:**
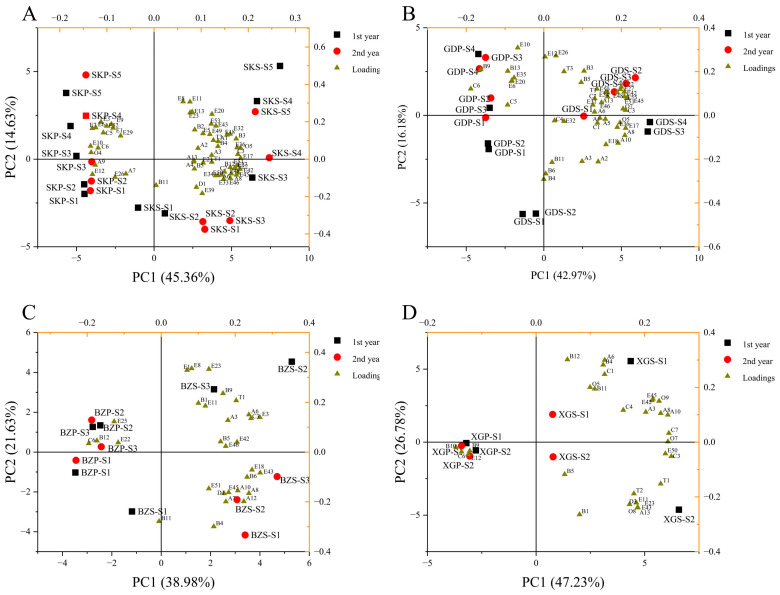
PCA of SK (**A**), GD (**B**), BZ (**C**), and XG (**D**) apple cultivars. Black squares represent samples from the first year, and red circles represent samples from the second year. Brown triangles represent the loadings of the VOCs. The codes of the VOCs are shown in Appendix A. SKS, GDS, BZS, XGS, and XGS represent skin samples from four apple cultivars, respectively; SKP, GDP, BZP, XGP, and XGP represent pulp samples from four apple cultivars, respectively. The same applies below.

**Figure 5 foods-13-02869-f005:**
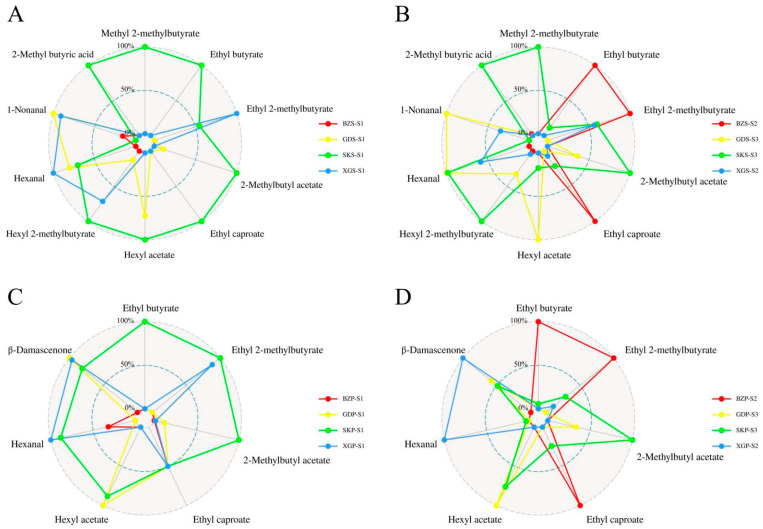
Radar map of the main characteristic VOCs of four apple cultivars in the first year. The radar maps at S1 in the skin (**A**) and pulp (**C**), respectively. The radar maps at the optimal aroma periods (SK at S3, GD at S3, BZ at S2, and XG at S2) in the skin (**B**) and pulp (**D**), respectively.

**Figure 6 foods-13-02869-f006:**
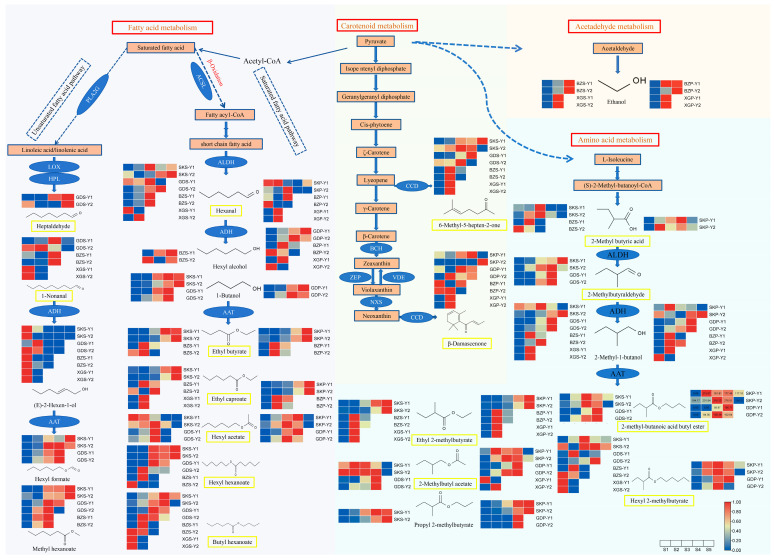
Heatmap of fatty acid, amino acid, carotenoid, and acetaldehyde metabolism pathways. Y1 and Y2 represent the first year and second year, respectively. The compounds in the yellow border represent characteristic aroma substances with high OAV. PLA2G: lipoprotein-associated phospholipase 2; ACSL: acyl-CoA synthetase; LOX: lipoxygenase; HPL: hydroperoxide lyase; ADH: alcohol dehydrogenase; ALDH: aldehyde dehydrogenase; AAT: alcohol acyltransferase; CCD: carotenoid cleavage dioxygenase; BCH: non-heme carotene hydroxylase; ZEP: zeaxanthin epoxidase; VDE: violaxanthin de-epoxidase; NXS: neoxanthin synthase.

**Table 1 foods-13-02869-t001:** The odor and OAV of the main characteristic VOCs in skin.

Compound	Aroma Descripition	Odor Threshold (µg/kg)	Fruit Harvest Day (S1)				Key Aroma Transition Period			
			SK	GD	BZ	XG	SK	GD	BZ	XG
Methyl 2-methylbutyrate	Apple, Fruit, Green Apple, Strawberry	0.25	1388.35	–	–	–	1679.36	–	–	–
Ethyl butyrate	Pineapple, fruity, apple	1	36.09	–	0.00	–	378.47	–	3439.60	–
Ethyl 2-methylbutyrate	Apple, Ester, Green Apple, Kiwi, Strawberry	0.1	332.13	–	0.00	608.27	9268.73	–	15,469.63	8740.38
2-Methylbutyl acetate	Apple, Banana, Pear	5	154.18	17.36	–	0.00	204.46	74.90	–	0.00
Ethyl caproate	Apple peel, fruit	1	153.26	–	0.00	0.00	249.09	–	1167.97	83.08
Hexyl acetate	Apple, Banana, Grass	2	227.07	164.92	–	–	92.73	532.83	–	–
Hexyl 2-methylbutyrate	Fruity, green, apple	6	103.91	12.97	0.00	74.47	527.09	183.89	19.27	41.42
Hexanal	Grass, green, leaves, vinous	4	282.97	320.70	23.32	392.64	402.54	407.84	42.80	257.22
1-Nonanal	Aldehyde, citrus, fatty, floral, green	1	–	166.74	26.49	151.52	–	335.04	0.00	116.24
2-Methyl butyric acid	Pungent, cheese, fruity	5.8	166.19	–	0.00	–	606.44	–	13.58	–

–: not detected. The key aroma transition periods of SK, GD, BZ, and XG were S3, S3, S2, and S2, respectively. The same applies below.

**Table 2 foods-13-02869-t002:** The odor and OAV of the main characteristic VOCs in pulp.

Compound	Aroma Descripition	Odor Threshold (µg/kg)	Fruit Harvest Day (S1)				Key Aroma Transition Period			
			SK	GD	BZ	XG	SK	GD	BZ	XG
Ethyl butyrate	Pineapple, fruity, apple	1	38.55	–	0.00	–	112.99	–	1993.95	–
Ethyl 2-methylbutyrate	Apple, Ester, Green Apple, Kiwi, Strawberry	0.1	171.95	–	0.00	151.44	2717.36	–	9328.17	1036.50
2-Methylbutyl acetate	Apple, Banana, Pear	5	148.21	18.16	–	2.06	247.94	83.06	–	0.00
Ethyl caproate	Apple peel, fruit	1	0.00	–	0.00	–	78.16	–	325.01	–
Hexyl acetate	Apple, Banana, Grass	2	85.07	96.45	–	–	317.65	417.48	–	–
Hexanal	Grass, green, leaves, vinous	4	10.64	–	3.87	12.05	4.54	–	0.00	136.53
β-Damascenone	Apple, rose, honey	0.05	691.43	795.87	241.95	775.81	769.58	921.84	0.00	1574.64

## Data Availability

The original contributions presented in the study are included in the article/Appendix A, further inquiries can be directed to the corresponding authors.

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
