# Peer review of "Analysis of Aroma Characteristics of ‘Binzi’ and ‘Xiangguo’ Apple—Ancient Cultivars in China"

_foods, 2024, doi:10.3390/foods13182869_

Round 1
Reviewer 1 Report
Comments and Suggestions for Authors
Please make note of the following points:
1. Specify the specific variety of Binzi that was used.
2. M domestica subsp. xiangguo: Please specify who identified this material and which publicly accessible and scientifically accepted document this subspecies is described, as it does not appear in the sources sought. Therefore, its characterization or the nomenclature used seems highly doubtful. Also, indicate which variety was used.
3. There are previous studies on Binzi.
4. In the tables, please indicate the Retention Index (RI) of each compound. Otherwise, it is impossible to determine if they were correctly identified using the mass spectrum alone.
5. Some boards have multiple sheets. Please indicate what each one is and always use the English language. If they are not necessary, they can be eliminated.
6. There are more compounds with their aromatic profiles published (smell and taste: see https://www.thegoodscentscompany.com/; doi:10.1021/ci0502505; etc.) and olfactory thresholds (numerous bibliographic references: ISBN/EAN: 978-90-810894-0-1; American Industry Hygiene Assoc. Journal (47) March, 1986; http://www.leffingwell.com/; etc.). The available information can be expanded to optimize results.
7. It is easier to read the results obtained in section 3.6 regarding flavor profiles if an Excel table is used, with the Odor Activity Value (OAV) data for each case.
Author Response
Point 1: Specify the specific variety of Binzi that was used.
Response 1: Binzi is an ancient cultivar in China, widely planted in Huailai County, Hebei Province. In the past few hundred years, throughout history, Binzi has been brought to other places by people, animals, or birds, such as Shandong, Shanxi, Gansu, and Beijing. In the process of natural selection, more than 20 types of Binzi have been produced.
In this study, the ‘Binzi’ is a unique cultivar discovered by our team during long-term field investigation, which is low-fragrant at harvest day and high-fragrant after brief storage at room temperature. The ‘Binzi’ has been preserved in our resource garden, and more types of Binzi are preserved. After years of observation, we have found that the aroma of ‘Binzi’ used in this study is the most unique. Therefore, we went to the discovery site for two consecutive years to collect and test for volatile compounds. These contents have been described in the introduction and method sections.
Point 2: M.domestica subsp. xiangguo: Please specify who identified this material and which publicly accessible and scientifically accepted document this subspecies is described, as it does not appear in the sources sought. Therefore, its characterization or the nomenclature used seems highly doubtful. Also, indicate which variety was used.
Response 2: ‘Xiangguo’ (Malus domestica subsp. chinensis var. xianggo Li. Y.N.) is named by Li Yunong (Li Y.N.). Li Yunnong, a fruit tree scientist and expert in apple germplasm resources, has been engaged in fruit tree teaching and research for a long time. He made significant contributions to the study of the origin, evolution, polymorphism, germplasm characteristics, classification, and distribution of apples. As the associate editor, he published the book "The Fruit Tree Encyclopedia: Apple Volume". More information about Xiangguo can be found in the book. The ‘Xiangguo’ used in this study is discovered by our team during a long-term field investigation. ‘Xiangguo’ is located in the same county as the ‘Binzi’ used in this study. These contents have been described in the introduction and method sections.
Point 3: There are previous studies on Binzi.
Response 3: The ‘Binzi’ used in this study were from Huailai County, Hebei Province, while previous studies were from Beijing (doi: 10.1186/s12870-022-03896-z), and they were not the same variety. Our results indicate that there are differences between them and have been clarified in the discussion.
Point 4: In the tables, please indicate the Retention Index (RI) of each compound. Otherwise, it is impossible to determine if they were correctly identified using the mass spectrum alone.
Response 4: Thank you for your valuable advice. We use similarity (SI) and reverse similarity (RSI) as references to manually compare each chromatographic peak and determine the final compound. This method can identify the correct VOCs. The detailed method is as follows:
The total ion flow chromatogram (TIC) was recorded in full scanning mode. The components of each chromatographic peak were identified by computer for matches in two mass spectrometry libraries (NIST and Wiley) and combined with artificial atlas and data. The mass spectra of all components under each chromatographic peak were compared with the standard mass spectra in the libraries to obtain similarity (SI) and reverse similarity (RSI), which were used to identify the component of each chromatographic peak. SI and RSI > 700 were the lowest reference standards. Meanwhile, previous studies on the composition of fruits have been reported as auxiliary references.
Point 5: Some boards have multiple sheets. Please indicate what each one is and always use the English language. If they are not necessary, they can be eliminated.
Response 5: We have made modifications to these contents.
Point 6: There are more compounds with their aromatic profiles published (smell and taste: see https://www.thegoodscentscompany.com/; doi:10.1021/ci0502505; etc.) and olfactory thresholds (numerous bibliographic references: ISBN/EAN: 978-90-810894-0-1; American Industry Hygiene Assoc. Journal (47) March, 1986; http://www.leffingwell.com/; etc.). The available information can be expanded to optimize results.
Response 6: Thanks for your suggestion. We have added more information about the compounds in Table S1 and made corresponding modifications. A total of 11 characteristic aroma components (OAV>1) have been added, and detailed information can be found in Table S4 and Table S5. The results of section 3.6 have been modified.
Point 7: It is easier to read the results obtained in section 3.6 regarding flavor profiles if an Excel table is used, with the Odor Activity Value (OAV) data for each case.
Response 7: Thanks for your suggestion. We have made modifications to these contents.
Point 8: Are the results clearly presented?
Response 8: We have made modifications to the results section.

Reviewer 2 Report
Comments and Suggestions for Authors
Authors
The manuscript ID foods-3181072 entitled “Analysis of Aroma Characteristics of ‘Binzi’ and 2 ‘Xiangguo’ Apple— Ancient Cultivars in China” in Foods proposes a very detailed description of the VOCs associated with the aroma of 4 different types of apples of Chinese origin. The paper is very rigorous and well written and justifies each of the claims made. The fact that it deals with 4 different types of apples makes the work very dense to read and sometimes complicated to follow, but it has the advantage of the large number of comparisons that the authors have been able to establish between the different types of apples. The work is also supported by a rigorous statistical analysis.
After careful consideration, this paper still needs some revisions.
Comments
Simplify the introduction by avoiding so much information about origin and giving more importance to the VOCs found in apples by other authors.The novelty of the method should be clearly stated in the introduction and the conclusion sections. The post-harvest storage conditions indicate that the apples were treated with liquid nitrogen until extraction, but in when were they treated? If all the samples were treated at the same time, do the apples follow the same ripening process as unfrozen apples?In section 2.2. replace sample inlet with GC inlet (line 115 and 118). Also, the ageing treatment is for the fiber and it is not clear in the text. In section 2.3. put the sub-indices in min-1 when temperature rates is referred. The calculation of the Odor activity value is not well understanding. Please, revise it. It is stated that a total of 105 compounds were detected. Does this represent the total volatile fraction of the apple or were many peaks not identified? What parameters and criteria were used for identification? The whole part of the GC-MS identification should be reviewed and supplemented with data. The quality of the figures are very low. The details in most of the figures cannot be seen. Please, revise the figures and increase the image quality.
It is not clear from Figure 1E which A-N sets are being referred to.
Author Response
Point 1: Simplify the introduction by avoiding so much information about origin and giving more importance to the VOCs found in apples by other authors.
Response 1: We have simplified the introduction of the cultivar and added more information on the VOCs from other authors.
Point 2: The novelty of the method should be clearly stated in the introduction and the conclusion sections.
Response 2: We have supplemented the novelty of the method.
Point 3: The post-harvest storage conditions indicate that the apples were treated with liquid nitrogen until extraction, but in when were they treated? If all the samples were treated at the same time, do the apples follow the same ripening process as unfrozen apples?
Response 3: Because of the different maturity periods of different apple cultivars, each apple cultivar was stored separately. After storage, We cut the fruits into pieces and treated them with liquid nitrogen, and stored them at -80 ℃. After all samples had been stored, we added NaCl and internal standard to the samples and placed them in an injection bottle; then detected the VOCs.
Point 4: In section 2.2. replace sample inlet with GC inlet (line 115 and 118). Also, the ageing treatment is for the fiber and it is not clear in the text. In section 2.3. put the sub-indices in min-1 when temperature rates is referred.
Response 4: The GC condition used in this study was consistent with our previous method. The detailed method was described in the paper (doi: 10.3390/agriculture12101710).
Point 5: The calculation of the Odor activity value is not well understanding. Please, revise it.
Response 5: We have added the calculation method for the odor activity value.
Point 6: It is stated that a total of 105 compounds were detected. Does this represent the total volatile fraction of the apple or were many peaks not identified?
Response 6: The 105 compounds detected in this study only represent the compounds of the 4 apple cultivars studied in this study. The 105 compounds can represent the vast majority of the peaks. There are still some peaks with extremely small areas that cannot be identified.
Point 7: What parameters and criteria were used for identification?
Response 7: We manually compared the compounds contained in each chromatographic peak, rather than having the instrument automatically export all the results. The mass spectra of all components under each chromatographic peak were compared with the standard mass spectra in the libraries to obtain similarity (SI) and reverse similarity (RSI), which were used to identify the component of each chromatographic peak. SI and RSI > 700 were the lowest reference standards. Meanwhile, previous studies on the composition of fruits have been reported as auxiliary references.
Point 8: The whole part of the GC-MS identification should be reviewed and supplemented with data.
Response 8: We have reviewed the part of the GC-MS identification and supplemented the data.
Point 9: The quality of the figures are very low. The details in most of the figures cannot be seen. Please, revise the figures and increase the image quality.
Response 9: We have revised the figures and increased the image quality.
Point 10: It is not clear from Figure 1E which A-N sets are being referred to.
Response 10: We have supplemented the information in Figure 1E. Set ‘A’ represented the common VOCs among the four apple cultivars; set ‘B’ represented the common VOCs between SK and GD; set ‘C’ represented the common VOCs among SK, GD, and XG; set ‘E’ represented the common VOCs between SK and BZ; set ‘F’ represented the common VOCs among SK, GD, and BZ; set ‘I’ represented the common VOCs between SK and XG; set ‘J’ represented the common VOCs among SK, BZ, and XG; set ‘L’ represented the common VOCs among GD, BZ, and XG; set ‘M’ represented the common VOCs between GD and XG; set ‘N’ represented the common VOCs between BZ and XG; set ‘D’, ‘G’, ‘H’, and ‘K’ represented the unique VOCs in SK, GD, BZ, and XG, respectively.
Point 10: Are the methods adequately described?
Response 10: We have made modifications to the method section.
